# AdaAug: Learning Class- and Instance-adaptive Data Augmentation Policies

**Tsz-Him Cheung & Dit-Yan Yeung**
Department of Computer Science and Engineering
The Hong Kong University of Science and Technology
{thcheungae, dyyeung}@cse.ust.hk

## Abstract

Data augmentation is an effective way to improve the generalization capability of modern deep learning models. However, the underlying augmentation methods mostly rely on handcrafted operations. Moreover, an augmentation policy useful to one dataset may not transfer well to other datasets. Therefore, Automated Data Augmentation (AutoDA) methods, like *AutoAugment* and *Population-based Augmentation*, have been proposed recently to automate the process of searching for optimal augmentation policies. However, the augmentation policies found are not adaptive to the dataset used, hindering the effectiveness of these AutoDA methods. In this paper, we propose a novel AutoDA method called `AdaAug` to efficiently learn adaptive augmentation policies in a class-dependent and potentially instance-dependent manner. Our experiments show that the adaptive augmentation policies learned by our method transfer well to unseen datasets such as the Oxford Flowers, Oxford-IIT Pets, FGVC Aircraft, and Stanford Cars datasets when compared with other AutoDA baselines. In addition, our method also achieves a state-of-the-art performance on the CIFAR-10, CIFAR-100, and SVHN datasets.[1]

## 1 Introduction

Data augmentation is a common way to enhance the robustness of deep learning models by augmenting the datasets used for model training. Applying popular data augmentation operations such as randomized cropping, horizontal flipping, and color shifting to image data has become a standard procedure in modern image recognition models (Krizhevsky et al., 2012; Shorten & Khoshgoftaar, 2019). Over the years, various augmentation methods using advanced operations have been proposed. Examples include occlusion-based operations like Cutout (Devries & Taylor, 2017) that randomly masks part of an image to avoid overfitting, label-mixing operations like CutMix (Yun et al., 2019) that replaces the occluded part in Cutout with a different image patch, and Mixup (Zhang et al., 2018) that interpolates two images with their corresponding one-hot encoded labels.

While these hand-crafted data augmentation methods can improve model generalization, choosing the operations and their corresponding parameters is often decided manually to make the augmentation scheme effective for the task at hand. Despite the manual efforts involved, an augmentation policy that is useful for a particular dataset does not often generalize well to other datasets (Cubuk et al., 2019). To tackle this problem, a series of recent studies has been conducted to automate the process of finding an effective data augmentation policy for a target dataset. These automated data augmentation (AutoDA) methodologies show impressive results on several benchmark image datasets (Cubuk et al., 2019; Ho et al., 2019; Lim et al., 2019; Cubuk et al., 2020; Hataya et al., 2020; Hendrycks et al., 2020; Li et al., 2020; Cheung & Yeung, 2021).

The manual tuning of augmentation parameters can also be addressed by using generative approaches, such as training a generative adversarial network (GAN) to create new artificial images directly (Antoniou et al., 2017; Tran et al., 2017; Jha & Cecotti, 2020; Yorioka et al., 2020; Zhao et al., 2020). However, these generative models are often hard to implement and computationally expensive to train.

---

[1]Code is available at https://github.com/jamestszhim/adaptive_augment

In supervised learning, data augmentation is considered to be a naive way to inject inductive biases, such as translational invariance, to a classifier. With the recent advances in representation learning, data augmentation has become a major approach to the learning of good representations. For example, there is an increasing trend to replace convolutional neural networks (CNNs) with transformers in computer vision (Carion et al., 2020; Dosovitskiy et al., 2021). Being a more generic architecture, transformers do not come with additional inductive bias such as translational invariance in CNNs, thereby requiring more data to learn the effective invariance properties. Data augmentation serves as an effective way to achieve this goal (Touvron et al., 2020). In addition, recent self-supervised models rely heavily on data augmentation to create different views of the same data and learn the robust representations through contrastive learning (Chen et al., 2020; Grill et al., 2020; He et al., 2020). However, an improper choice of augmentation operations, especially with excessive strength, may impose a wrong inductive bias to the models and lead to performance degradation (Chen et al., 2020; Xiao et al., 2021). Consequently, there is a need to enrich the current data augmentation methods, especially with a better choice of the data augmentation policy, to take advantage of the recent advances in representation learning.

Previous AutoDA methods attempt to find an optimal augmentation policy to augment a given dataset. However, most of the discovered policies are not adaptive to variations of the dataset. Even in a dataset in which an augmentation policy is found to be effective, the same augmentation scheme is applied equally to all classes. This can limit the potential of data diversity brought about by data augmentation. For example, in digit classification, flip-invariance is useful for the digits "0", "1" and "8" but not for the other digits; in shape classification, shearing-invariance is useful for "triangles" but not for "rectangles". None of the previous AutoDA methods can learn an adaptive class-dependent augmentation policy. To address this limitation, we propose `AdaAug`, as an AutoDA method, to learn a class-dependent and potentially instance-dependent augmentation policy efficiently.

Despite the attractive potential of such an adaptive scheme if it can indeed be realized, actually learning an adaptive augmentation policy poses at least two major technical challenges. First, the search space for the per-class, per-instance augmentation policies is very large, rendering it intractable when trying to maintain an individual policy for each class or even each realization of the input data. Second, the gradient information of the augmentation parameters is hard to obtain as the operation selection process and the transformations are non-differentiable. Optimizing the augmentation policy efficiently is a challenging problem to address. In this work, `AdaAug` employs a recognition model to learn the underlying augmentation policy for each data instance and takes an alternating exploit-and-explore procedure to update the augmentation policy using a differentiable workflow. In the exploitation pass, `AdaAug` trains a classifier for a number of steps, followed by the exploration pass which validates the classifier and updates the policy to minimize the validation loss. The intuition behind our design of `AdaAug` is that such an alternating procedure can learn augmentations that help generalize the trained model to unseen validation data. For example, rotational invariance would be found to be a desirable inductive bias that the model should learn if the validation set contains some similar but rotated versions of the training images. Our goal is to capture such information and assign a higher probability to use rotation for augmentation in this case. An application scenario would be a computer vision task for drones where the unseen data may contain different kinds of rotated images. To summarize, our contributions are listed as follows:

- We introduce a novel AutoDA method to learn a class-dependent and potentially instance-dependent augmentation policy for each data instance.

- We propose a differentiable workflow to search for the augmentation policy efficiently.

- We demonstrate that the policies learned by our method transfer better to unseen datasets, such as Oxford Flowers, Oxford-IIT Pets, FGVC Aircraft, and Stanford Cars, when compared to other AutoDA baselines.

- We demonstrate a state-of-the-art performance on the CIFAR-10, CIFAR-100, and SVHN datasets.

## 2 RELATED WORK

**Automated Data Augmentation.** Several AutoDA methods have been proposed to compose the augmentation operations automatically. AutoAugment (Cubuk et al., 2019) learns to generate the probability and magnitude when applying different augmentation operations as a policy using reinforcement learning (RL). It alternately generates an augmentation policy to train a child model and updates the policy generator using the validation performance as a reward. Since it is computationally expensive to train the child models repeatedly, several techniques have been proposed subsequently in an attempt to reduce the search effort. Fast AutoAugment (Lim et al., 2019) uses Bayesian optimization (BO) to tune the augmentation parameters. Population-based Augmentation (PBA) (Ho et al., 2019) exploits population-based training (PBT) to search for an optimal augmentation policy schedule by training multiple parallel child models using an evolutionary approach. RandAugment (Cubuk et al., 2020) applies the augmentation operations uniformly and reduces the search space significantly by covering only the number of operators and the global augmentation magnitude. Instead of using the validation performance to evaluate the augmentation quality, Adversarial AutoAugment (Zhang et al., 2020) uses an adversarial objective to learn the augmentation policy. MODALS (Cheung & Yeung, 2021) utilizes PBA to search for an optimal latent space augmentation policy and augment data from any modality not limited to image data.

**Differentiable Data Augmentation.** In addition to using RL, BO and PBT to optimize the augmentation parameters, there exist related methods that modify the otherwise discrete search procedures to make them end-to-end differentiable. This results in a more efficient optimization procedure and a more precise policy than RL and PBT as the search space is continuous. In `AdaAug`, part of our contribution is to design a differentiable workflow to learn the augmentation policy. For previous differentiable augmentation approaches, Faster AutoAugment (Hataya et al., 2020) proposes a differentiable Relaxed Bernoulli distribution to sample the candidate augmentation functions and estimate the gradients of the non-differentiable augmentation magnitude using the Stop Gradient estimator. Specifically, it optimizes a density matching loss between the training and validation data. DADA (Li et al., 2020) differentiates through the discrete policy sampling process using Gumbel-Softmax trick. While AutoDA and differentiable data augmentation have been shown to be successful in improving the generalization performance of deep learning models, the learned augmentation policy is often applied uniformly to the whole dataset, meaning that all classes and instances share the same augmentation policy. On the contrary, each class and even each data instance receives an adaptive policy to augment in our proposed method.

**Adaptive Data Augmentation.** Attempts have been made to apply adaptive data augmentation at a class or subgroup level. Hauberg et al. (2016) model the transformations within each class and use statistical models to augment the dataset. The approach shows improvement in the small MNIST dataset and its variants. However, the augmentation operations are limited to spatial transformations. In addition, observations of the data must be locatable and alignable, making it difficult to extend to most other computer vision tasks. Recently, CAMEL (Goel et al., 2021) adopts a finer data-generation method by fixing the classifiers that fail on a subgroup of a class. It uses CycleGAN to learn different variations of the same training data within a subgroup. However, CAMEL requires the specifying of the subgroup information manually and assumes that the subgroups only exist within the same class. MetaAugment (Zhou et al., 2021) learns a sample-wise weighting scheme and a global probability parameter to control the sampling of augmentation transformations. In `AdaAug`, the learned policy can capture class-dependent transformation automatically and also instance-dependent information, such as the light intensity of an image, across different classes.

## 3 ADAAUG

### 3.1 SEARCH SPACE

Let $\mathbb{T}$ be a set of augmentation operations where $\tau_j$ denotes the $j$-th operation (e.g., "rotation") in the set. We formulate an augmentation policy as the probability $p$ and magnitude $\lambda$ in applying the augmentation operations. Here, $p$ is a probability vector with each entry $p_j \in [0, 1]; \sum_{j=1}^{|\mathbb{T}|} p_j = 1$ and $\lambda$ is a vector with each entry $\lambda_j \in [0, 1]$, where $p_j$ and $\lambda_j$ are the probability and magnitude, respectively, of applying the operation $\tau_j$. Mathematically, $\tau_j : \mathcal{X} \rightarrow \mathcal{X}$ is a mapping from the input space $\mathcal{X}$ to itself. For an image $x \in \mathcal{X}$, $\tau_j$ transforms it with the magnitude parameter $\lambda_j$ that

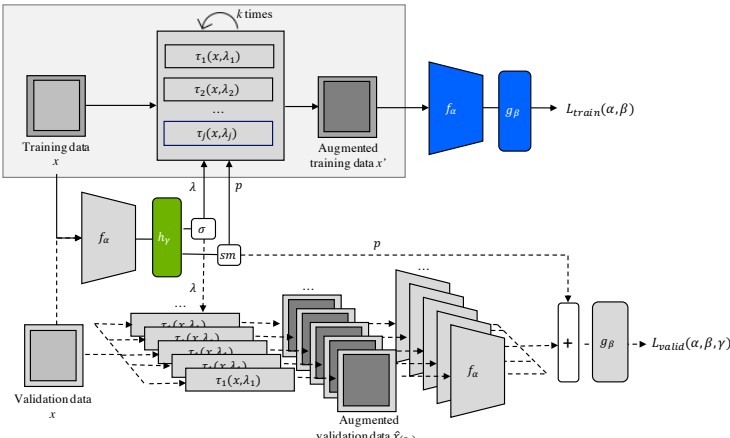

Figure 1: Illustration of the `AdaAug` pipeline. In the exploitation pass (solid line), the training images are augmented by the policy network ($h_\gamma \circ f_\alpha(x)$) as new unseen data to train the classifier (colored in blue). In the exploration pass (dotted line), each validation data instance passes through multiple augmentation paths. The corresponding latent representations are summed with the generated augmentation probabilities as weights. The policy projection network (colored in green) is updated to minimize the validation loss. To compute the probability and magnitude vectors, $\sigma$ and $sm$ denote the sigmoid and softmax functions, respectively.

specifies the strength of the transformation (e.g., degree of rotation), i.e., $x \mapsto \tau_j(x; \lambda_j)$. Note that some operations like flipping do not depend on the magnitude parameter. In a training pass, given an input data $x$, an augmentation policy $(p, \lambda)$, and the number of operators $k$, we sample $k$ operations according to $p$ and apply them with their corresponding magnitudes specified by $\lambda$:

$$
\begin{aligned}
\mathcal{T}(x; p, \lambda) &= \tau_j(x; \lambda_j); \ j \sim p \\
\hat{x} &= \mathcal{T}^{(k)} \circ \cdots \circ \mathcal{T}^{(1)}(x; p, \lambda)
\end{aligned}
\tag{1}
$$

Here, $\mathcal{T}^{(t)}, 1 \le t \le k$, denotes applying the $t$-th operation. Our goal is to learn an augmentation policy function $\pi_\theta : x \mapsto (p, \lambda)$ to generate an adaptive augmentation policy that optimizes the generalization performance and is dependent on the input data (see Figure 1 and Algorithm 1).

## 3.2 SEARCH ALGORITHM

**Exploitation.** `AdaAug` uses a feature extraction network $f_\alpha : \mathcal{X} \to \mathcal{Z}$ to map an input space to a latent space, a dense layer $g_\beta : \mathcal{Z} \to \mathcal{Y}$ to map a latent space to a label space, and a projection function $h_\gamma : \mathcal{Z} \to P \times \Lambda$ to map a latent space (representation) to a probability and magnitude space, where $p = [0, 1]^{|\mathbb{T}|}$ and $\|p\|_1 = 1, \forall p \in P$ and $\lambda = [0, 1]^{|\mathbb{T}|}, \forall \lambda \in \Lambda$. In our case, the functions $f, g, h$ (with the subscripts dropped for notational simplicity) are implemented as neural networks with network weights $\alpha, \beta, \gamma$, respectively. The softmax operation is applied to the first half of the output from $h$ to get the probabilities and the sigmoid function is applied to the other half to get the magnitudes. The policy network $\pi_\theta(x)$ can then connect the class and image information to the augmentation policy via $h \circ f(x)$ with parameters $\theta = (\gamma, \alpha)$. In an exploitation pass, given the training data $x$, the policy function generates the data-dependent augmentation policy $(p, \lambda) = \pi_\theta(x)$ and augments $x$ to give $\hat{x}$ using Equation (1). Here, $\hat{x}$ is treated as a new unseen training example and is used to train the classification model $g \circ f(\hat{x})$ by minimizing the cross-entropy loss: $\min_{\alpha, \beta} \mathcal{L}_{train}(\alpha, \beta)$. Note that there is a discrete sampling procedure in Equation (1). During the update, the gradient of $\alpha$ does not involve the computations in the policy network.

**Exploration.** In the exploration pass, `AdaAug` generates the augmentation policy $(p, \lambda) = \pi_\theta(x)$ given the validation data $x$. The $|\mathbb{T}|$ operations are applied to $x$ separately with the corresponding magnitudes in $\lambda$. The augmented validation data are passed to the feature extraction network $f$ individually to get the latent representations, which are then summed based on their weights in the

probability vector $p$. The mixed representation is passed to $g$ for computing the predicted labels:

$$\hat{y} = g\left(\sum_{j=1}^{|\mathbb{T}|} p_j \cdot f \circ \tau_j(x; \lambda_j)\right); \qquad (p, \lambda) = \pi_\theta(x) \qquad (2)$$

$\texttt{AdaAug}$ updates $\gamma$ to minimize the validation loss: $\min_\gamma \mathcal{L}_{valid}(\alpha, \beta, \gamma)$. As we make no assumption that the augmentation operations are differentiable, we follow prior approaches (Bengio et al., 2013; Li et al., 2020) using a straight-through gradient estimator to optimize the augmentation magnitudes. Specifically, the gradient of the magnitudes is estimated with respect to each input pixel value $x_{h,w}$ of the augmented data, i.e., $\frac{\partial \hat{x}_{h,w}}{\partial \lambda_j} = 1$. The gradient can then be calculated by:

$$\frac{\partial \mathcal{L}_{valid}}{\partial \lambda_j} = \sum_{w,h} \frac{\partial \mathcal{L}_{valid}}{\partial \hat{x}_{w,h}} \frac{\partial \hat{x}_{w,h}}{\partial \lambda_j} = \sum_{w,h} \frac{\partial \mathcal{L}_{valid}}{\partial \hat{x}_{w,h}} \qquad (3)$$

---

**Algorithm 1** Search algorithm

---

1: **procedure** SEARCH($D_{train}, D_{valid}, \mathbb{T}, k, m, r$)
2:    Initialize $\alpha, \beta, \gamma$
3:    **for** $r$ steps **do** *# Exploitation*
4:        Sample a mini-batch $d_{train} \in D_{train}$
5:        **for** $(x, y) \in d_{train}$ **do**
6:            $(p, \lambda) = \pi_\theta(x)$
7:            $\hat{x} = \texttt{augment}(x)$                                  ▷ $\texttt{augment}$ computes Eq. 1
8:            $\hat{y} = g \circ f(\hat{x})$
9:            $\mathcal{L}_{train}(\alpha, \beta) = CrossEntropyLoss(\hat{y}, y)$
10:           $(\alpha, \beta) \leftarrow \arg\min_{\alpha,\beta} \mathcal{L}_{train}(\alpha, \beta)$
11:       **if** $r$ is divisible by $m$ **then** *# Exploration*
12:           Sample a mini-batch $d_{valid} \in D_{valid}$
13:           **for** $(x, y) \in d_{valid}$ **do**
14:               $(p, \lambda) = \pi_\theta(x)$
15:               $\hat{y} = \texttt{explore}(x)$                              ▷ $\texttt{explore}$ computes Eq. 2
16:               $\mathcal{L}_{valid}(\alpha, \beta, \gamma) = CrossEntropyLoss(\hat{y}, y)$
17:               $\gamma \leftarrow \arg\min_\gamma \mathcal{L}_{valid}(\alpha, \beta, \gamma)$
       **return** $\alpha, \gamma$

---

**Relation to Neural Architecture Search (NAS).** Tuning of the augmentation policy parameters bears some similarity to the optimization of the network architecture weights in DARTS from the NAS literature (Liu et al., 2019). DARTS prescribes different computation paths to different operation cells and relaxes the computation to a mixture of the operations weighted by learnable weights for each path. With $w$ as the model parameters and $\alpha$ as the weights for the computation paths, the search algorithm solves a bi-level optimization problem:

$$\min_\alpha \mathcal{L}_{valid}(w^*(\alpha), \alpha) \quad \text{s.t.} \quad w^*(\alpha) = \arg\min_w \mathcal{L}_{train}(w, \alpha) \qquad (4)$$

While DARTS optimizes the architecture weights, $\texttt{AdaAug}$ optimizes the projection parameter $\gamma$ that decides the augmentation weights and magnitudes in the exploration. DARTS solves the optimization by a first-order and finite-difference approximation of the architecture gradient. $\texttt{AdaAug}$ samples the augmentation operations and treats the augmented data as missing training data $\hat{X}_{train}$. By absorbing the augmentation parameter in the training dataset, we avoid the complex bi-level optimization and simplify the exploitation procedure into training a standard classifier:

$$\min_\gamma \mathcal{L}_{valid}(\alpha^*, \beta^*, \gamma; X_{valid}) \quad \text{s.t.} \quad \alpha^*, \beta^* = \arg\min_{\alpha,\beta} \mathcal{L}_{train}(\alpha, \beta; \hat{X}_{train}) \qquad (5)$$

**Relation to Density Matching.** Data augmentation can be regarded as a density matching problem between the training and validation data (Ratner et al., 2017; Tran et al., 2017; Hataya et al., 2020; Lim et al., 2019). From this perspective, $\texttt{AdaAug}$ improves model generalization by matching the density of $D_{train}$ with the density of the augmented $D_{valid}$. In the outer optimization objective in Equation (5), $\texttt{AdaAug}$ minimizes the classification loss with the augmentation parameters $\gamma$ over the same optimal model parameters $\alpha^*, \beta^*$ learned from $\hat{D}_{train}$. In so doing, it approximately reduces the distance between the densities of the augmented $D_{valid}$ and $D_{train}$.

### 3.3 INFERENCE

Like most automated augmentation pipelines (Cubuk et al., 2019; Ho et al., 2019; Lim et al., 2019; Hataya et al., 2020; Li et al., 2020), `AdaAug` searches for the augmentation policy on a small dataset using a small model and then applies the learned policy network $\pi_\theta$ to train with a larger dataset or model. We call the process of applying a searched policy to augment a new dataset as inference time. Regarding this workflow, RandAugment argues that the different search spaces in search time and inference time makes the augmentation policy unable to adjust the regularization strength to different target datasets and models (Cubuk et al., 2020). Therefore, it proposes to search for the global magnitude and number of operators for each case. To address this concern, `AdaAug` utilizes three diversity parameters to fine-tune the regularization strength of the found policy for large datasets and models. First, the number of operators $k$ is set to 1 at search time, but a larger value of $k$ can be used at inference time. To control the selection of more diverse operations, a temperature parameter $T$ is introduced in the softmax function: $p_i = \frac{\exp\left[h(z_i)/T\right]}{\sum_j^{|\mathbb{T}|} \exp\left[h(z_j)/T\right]}$. Setting a larger $T$ value allows the operations with lower probability being sampled more often in the same order. Last, `AdaAug` perturbs the magnitude value with the parameter $\delta$ so the perturbed magnitude $\hat{\lambda}$ is given as $\hat{\lambda} \sim Uniform(\lambda - \delta, \lambda + \delta)$. This allows the augmented data to show slight variations even with the same operation on the same input data. In practice, we can perform grid search on these three diversity parameters with the other hyperparameters using a holdout validation set.

## 4 EXPERIMENTS AND RESULTS

In this section, we explain our experimental design followed by presenting the results. We evaluate the empirical performance of `AdaAug` in two experiments: AdaAug-transfer and AdaAug-direct. We select the comparison baselines that use the validation performance as the evaluation method to learn the augmentation policy. For all the experiments, we report the average test-set error rate as the performance metric. Each model is evaluated three times with different random initializations.

**Augmentation operations.** We match the operations adopted by AutoAugment. In addition to the 16 operations proposed previously (ShearX, ShearY, TranslateX, TranslateY, Rotate, AutoContrast, Invert, Equalize, Solarize, Posterize, Contrast, Color, Brightness, Sharpness, Cutout, and Sample Pairing), we add the Identity operation for not applying data augmentation. For the simple baseline, we apply random horizontal flip, color jittering, color normalization, and Cutout with a $16 \times 16$ patch size. Our method and other baselines apply the found policy on top of these standard augmentations.

**Policy search.** We follow the setup adopted by AutoAugment (Cubuk et al., 2019) to use 4,000 training images for CIFAR-10 and CIFAR-100, and 1,000 training images for SVHN. The remaining images are used as the validation set. We use Wide-ResNet-40-2 (Zagoruyko & Komodakis, 2016) as the feature extraction network for all searches. We implement $h$ as a linear layer and update the policy parameter $\gamma$ after every 10 training steps using the Adam optimizer with a learning rate of 0.001 and a batch size of 128.

**AdaAug-transfer.** In the first experiment, we investigate how well the learned augmentation policy can transfer to unseen datasets. We search for the optimal augmentation policy on the CIFAR-100 dataset and use the learned policy to train with four fine-grained classification datasets: Oxford 102 Flowers (Nilsback & Zisserman, 2008), Oxford-IIIT Pets (Em et al., 2017), FGVC Aircraft (Maji et al., 2013), and Stanford Cars (Krause et al., 2013). We compare the test error rate with AutoAugment, Fast AutoAugment, DADA, and RandAugment using their published policies for CIFAR-100. For all the tested datasets, we compare the transfer results when training the ResNet-50 model (He et al., 2016) for 180 epochs from scratch and fine-tuning the ResNet-50 model pretrained on ImageNet for 100 epochs. We use the cosine learning rate decay with one annealing cycle (Loshchilov & Hutter, 2017), initial learning rate of 0.1, weight decay 1e-4 and gradient clipping parameter 5.

**AdaAug-direct**. In the second experiment, we search for the optimal augmentation policy on a small subset of the target dataset and use the learned policy to train the full dataset. The purpose of the experiment is to demonstrate that while the `AdaAug` policy can adapt to other unseen datasets, it can also achieve competitive performances on the seen datasets with more training data. We compare AdaAug-direct with state-of-the-art AutoDA methods using the same evaluation datasets: CIFAR-10, CIFAR-100 (Krizhevsky & Hinton, 2009), and SVHN (Netzer et al., 2011). We test our method

Table 1: Test error (%) when training ResNet-50 with different augmentation methods.

| Dataset | Train Size | Classes | Simple | AA | Fast AA | DADA | RA | AdaAug |
|---|---|---|---|---|---|---|---|---|
| Oxford 102 Flowers | 6,552 | 102 | 5.87 | 3.71 | 3.99 | 5.06 | 4.28 | **3.63 $\pm$ 0.14** |
| Oxford-IIIT Pets | 3,680 | 37 | 26.42 | 22.61 | 22.11 | 23.80 | 27.84 | **20.14 $\pm$ 0.12** |
| FGVC Aircraft | 6,667 | 100 | 19.87 | 17.72 | 17.80 | 18.84 | 17.70 | **17.50 $\pm$ 0.19** |
| Stanford Cars | 8,144 | 196 | 13.64 | 11.96 | 12.81 | 12.86 | 12.21 | **11.51 $\pm$ 0.21** |

Table 2: Test error (%) when fine-tuning pretrained ResNet-50 with different augmentation methods.

| Dataset | Simple | AA | RA | Fast AA | AdaAug |
|---|---|---|---|---|---|
| Oxford 102 Flowers | 5.02 | 6.12 | 4.77 | 3.92 | **2.81 $\pm$ 0.13** |
| Oxford-IIIT Pets | 19.47 | 18.75 | 23.03 | 16.79 | **16.14 $\pm$ 0.07** |
| FGVC Aircraft | 18.40 | 16.61 | 17.02 | 17.44 | **16.03 $\pm$ 0.15** |
| Stanford Cars | 11.92 | 9.18 | 10.72 | 10.29 | **8.82 $\pm$ 0.08** |

using Wide-ResNet-40-2 and Wide-ResNet-28-10. At inference time, we set the temperature $T = 3$ and magnitude perturbation $\delta = 0.3$ and search for the number of operators $k \in \{1, 2, 3, 4\}$ using a holdout validation set, like RandAugment (Cubuk et al., 2020). For the hyperparameters, we follow AutoAugment, PBA, and Fast AutoAugment if possible.

## 4.1 ADAAUG-TRANSFER

Table 1 and Table 2 show that the AdaAug policy outperforms the other baselines when training and fine-tuning the ResNet-50 model on the Flowers, Pets, Aircraft, and Cars datasets, respectively. These baselines apply the same augmentation policy to all datasets. Such a policy may not be optimal to the target domain. In contrast, AdaAug adapts the augmentation policy to individual image classes and instances automatically. The AdaAug policy network applies different augmentation policies to unseen images according to their similarity to the classes that AdaAug has seen in the search. To further justify our claim, we show the distribution of the augmentation parameters for different tasks in Appendix A.5.4. The four datasets have many classes but few training examples per class. The negative effect of using a non-adaptive data augmentation policy is likely to be more imminent than a situation where its dataset has fewer classes but more examples per class.

## 4.2 ADAAUG-DIRECT

**CIFAR-10 and CIFAR-100**. The policies learned by AdaAug mostly achieve either comparable or better performances than the baselines for both WRN-40-2 and WRN-28-10 models on the CIFAR-10 and CIFAR-100 datasets (see Table 3). We visualize the augmentation policy of CIFAR-10 by applying the policy to the validation data and averaging the predicted augmentation probability of each image for each class (see Figure 2). The policy contains all types of augmentations at some degree. Among the operations, Flip dominates the policy. It is in line with the augmentation selected manually as people find applying horizontal flipping to CIFAR-10 can improve the prediction accuracy. The policy learned by AdaAug puts higher emphasis on Invert and Equalize, which are also reported by PBA. The focus on Brightness is also aligned with the policy in AutoAugment. Although there are minor variations in the importance of some operations between the policies learned by AdaAug, AutoAugment, and PBA, their empirical performance is similar.

**SVHN**. AdaAug performs comparably to the baselines on the core set of SVHN. We visualize the augmentation policy for SVHN in Figure 2. We found that AutoContrast, Invert, and Solarize receive much attention in SVHN. This makes sense because the specific color of the number and background is irrelevant to the prediction. This is consistent with the findings by AutoAugment and PBA. The Flip operation receives a significantly higher probability in digits "0", "1", and "8". It is likely because these three digits appear similar after flipping. This shows that AdaAug captures not only a dataset-specific augmentation policy, but also class-dependent augmentation, which cannot be achieved by other baselines. In addition to the augmentation probability, we visualize the learned augmentation magnitude in Appendix A.5.2.

Table 3: Test error (%) on Reduced CIFAR-10, CIFAR-10, CIFAR-100, reduced SVHN and the SVHN core set ($^*$evaluated using the policy released by the baseline).

| | Simple | AA | PBA | Fast AA | DADA | Faster AA | RA | AdaAug |
|---|---|---|---|---|---|---|---|---|
| **Reduced CIFAR-10** | | | | | | | | |
| WRN-28-10 | 17.1 | 14.1 | **12.8** | 14.6$^*$ | 15.6$^*$ | - | 15.1$^*$ | 13.6 ± 0.03 |
| **CIFAR-10** | | | | | | | | |
| WRN-40-2 | 5.3 | 3.7 | 3.9$^*$ | **3.6** | **3.6** | 3.7 | 4.1$^*$ | **3.6 ± 0.11** |
| WRN-28-10 | 3.9 | 2.6 | **2.6** | 2.7 | 2.7 | **2.6** | 2.7 | **2.6 ± 0.10** |
| **CIFAR-100** | | | | | | | | |
| WRN-40-2 | 26.0 | 20.6 | 22.3$^*$ | 20.7 | 20.9 | 21.4 | - | **19.8 ± 0.08** |
| WRN-28-10 | 18.8 | 17.1 | **16.7** | 17.3 | 17.5 | 17.3 | **16.7** | 17.1 ± 0.12 |
| **Reduced SVHN** | | | | | | | | |
| WRN-28-10 | 13.2 | 8.2 | 7.8 | 8.1$^*$ | **7.6**$^*$ | - | 9.4$^*$ | 8.2 ± 0.12 |
| **SVHN (core set)** | | | | | | | | |
| WRN-28-10 | 3.1 | 1.9 | 2.1$^*$ | 2.3$^*$ | 2.4$^*$ | - | **1.7** | 2.3 ± 0.13 |

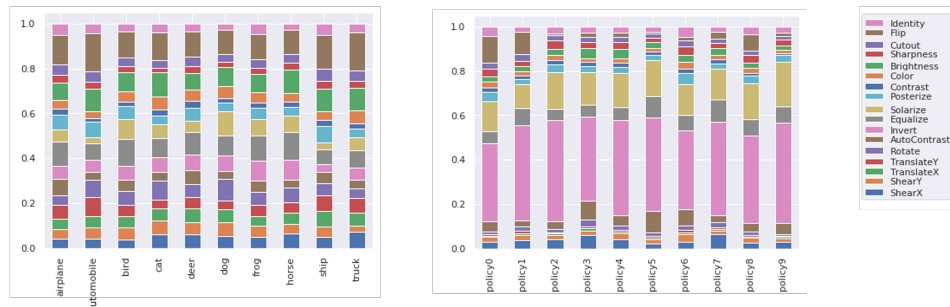

Figure 2: Illustration of the learned augmentation probability for CIFAR-10 (left) and SVHN (right).

**ImageNet**. We also validate our method on large-scale ImageNet dataset. `AdaAug` improves the top-1 accuracy 1% over the ResNet-50 baseline (see Appendix A.1.1). Although some baselines like AutoAugment produce similar performance or slightly outperform our method in the AdaAug-direct experiment, `AdaAug` uses less computational effort in searching for the policy. Specifically, AutoAugment takes 5,000 GPU hours to search for the CIFAR-10 policy, while `AdaAug` takes only 3.3 GPU hours on an old GeForce GTX 1080 GPU card (see Appendix A.4).

## 5 DISCUSSION

**Instance-dependent augmentation.** The AdaAug-direct results show that the learned `AdaAug` policy can capture dataset- and class-dependent augmentations on CIFAR-10, CIFAR-100, and SVHN. We further investigate whether `AdaAug` can learn instance-dependent information. According to the architecture, `AdaAug` takes the output from the last layer of a CNN network as the image representation and uses it to predict the augmentation parameters. The image representation contains the class information and potentially some image features. Empirically, we first examine if `AdaAug` generates different augmentation policies for different instances even within the same class. In Appendix A.5.3, we plot the standard deviations of the predicted augmentation probabilities of the image instances for each class. It is observed that even within the same class, the predicted augmentation policy is slightly different across instances. It is a clue that the `AdaAug` policy captures some instance-level augmentation information. Qualitatively, we show the augmented examples of a flower image using the `AdaAug` policy in Figure 3. The input image is relatively darker than the other images. Apparently, `AdaAug` is aware of this property and applies more brightness-related augmentation to lighten up the image. We observe similar behaviour of `AdaAug` in other classes, but there is inadequate empirical support to conclude a general rule on how the network decides the instance-aware augmentation as policies are learned in a data-driven way.

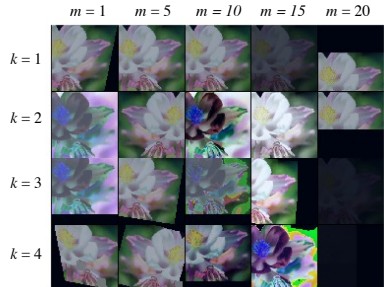 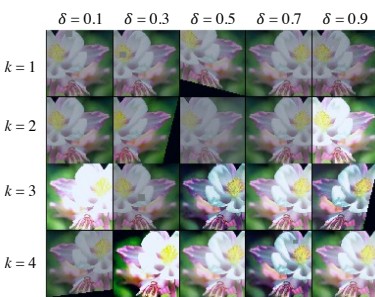

Figure 3: Visualization of the augmented images under different augmentation strengths using: RandAugment (left); and `AdaAug` (right). ($k$ is the number of operators, $m$ is the magnitude parameter in RandAugment, and $\delta$ is the magnitude perturbation parameter in `AdaAug`.)

**Quality of augmented data.** We compare the augmented images from `AdaAug` with RandAugment under different augmentation strengths in Figure 3. In terms of augmentation diversity, RandAugment produces more variations of the input image. However, not all augmented images produced are plausible. As RandAugment applies the augmentations uniformly, some augmented flower images show a strange color difference to the original image. With an increasing number of operators and magnitude, the flower object is sometimes translated out of the frame and results in a black image. This may affect the learning performance as color can be an important feature in classifying different species of flowers. For `AdaAug`, the augmented images are more visually appealing.

**Ablation study.** In this section, we study the effects of AdaAug-transfer on Oxford 102 Flowers and AdaAug-direct on CIFAR-10 using three alternative search configurations. First, does the use of a nonlinear projection deliver a better performance? We replace the linear layer in $h$ with a 2-layer MLP using a 128 hidden layer size with ReLU activation. Second, we study whether the class-dependent and instance-dependent configuration improves model accuracy. We remove the class and instance information by replacing the policy network with a fixed vector, which decides the augmentation probabilities and magnitudes for the entire dataset. In the third setting, we mix the augmented images in the input space instead of the latent space and observe the effects. Our results show that the use of class- and instance-adaptive augmentation policies contribute larger improvements in AdaAug-direct. In AdaAug-transfer, using a nonlinear projection harms the prediction performance. A possible reason is that the nonlinear projection is more likely to overfit the search dataset and fail to generalize to unseen datasets. Moreover, combining the augmentation path in the latent space learns a better policy (see Table 4).

Table 4: Test error rate (%) on AdaAug-transfer and AdaAug-direct with different search settings.

|          | `AdaAug` | `AdaAug` w/ 2-layer MLP as the projection function | `AdaAug` w/o class, instance information | `AdaAug` w/ input-space mixing |
|----------|----------|---------------------------------------------------|------------------------------------------|--------------------------------|
| transfer | **3.7**  | 4.5                                               | 3.9                                      | 4.0                            |
| direct   | **3.6**  | 3.7                                               | 4.4                                      | 3.9                            |

## 6 CONCLUSION

In this work, we propose a novel AutoDA approach, `AdaAug`, to learn class- and instance-adaptive augmentation policies efficiently. We demonstrate that the found policy transfers well to unseen datasets while achieving state-of-the-art results on the seen datasets. We provide evidence to show that the learned adaptive policy captures class- and instance-level information. We think that `AdaAug` can show further gains over existing baselines when applied to datasets with more dissimilar underlying augmentation rules among the data classes, and with fewer training examples for each class. It is also promising to investigate in the future if the proposed adaptive augmentation method can improve the performance of other computer vision and representation learning tasks.

ACKNOWLEDGMENTS

This research has been made possible by the Hong Kong PhD Fellowship provided to the first author and research grants (General Research Fund project 16204720 and Collaborative Research Fund project C6030-18GF from the Research Grants Council of Hong Kong; Amazon Web Services Machine Learning Research Award 2020) provided to the second author.

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

## A    APPENDIX

### A.1    ADDITIONAL EXPERIMENTS

#### A.1.1    LARGE-SCALE DATASET

For large-scale dataset, `AdaAug` improves the top-1 accuracy 1% over the ImageNet ResNet-50 baseline (see Table 5). The performance gain is similar to previous AutoDA methods and validates the positive effect of `AdaAug` on complex dataset.

#### A.1.2    ARCHITECTURE TRANSFER

We also provide the experimental results when using the learned Augmentation policy to train the Shake-Shake (26 2x96d) model on Reduced CIFAR-10 and Reduced SVHN in Table 6.

Table 5: Test error rates (%) on ImageNet using ResNet-50.

| Simple | AA | Fast AA | DADA | Faster AA | RA | AdaAug |
|--------|------|---------|------|-----------|------|--------|
| 23.7 | 22.4 | 22.4 | 22.5 | 23.5 | 22.4 | 22.8 |

Table 6: Test error rates (%) of Shake-Shake (26 2x96d) on Reduced CIFAR-10 and reduced SVHN

| | Simple | AutoAugment | PBA | AdaAug |
|---|--------|-------------|------|--------|
| **Reduced CIFAR-10** Shake-Shake (26 2x96d) | 17.05 | 10.04 | 10.64 | $10.92 \pm 0.07$ |
| **Reduced SVHN** Shake-Shake (26 2x96d) | 13.32 | 5.92 | 6.46 | $6.44 \pm 0.17$ |

## A.2 Additional Ablation Study

To clarify the improvements of AdaAug, we compare the performance of AdaAug under different settings in Table 7: *Simple*: standard data augmentation is applied; *Random*: AdaAug is applied with randomly initialized $h_\gamma$ while keeping the diversity parameters the same; *AdaAug (w/o diversity)*: AdaAug is applied without the diversity parameters; *AdaAug*: AdaAug is applied with learned $h_\gamma$ and the diversity parameters.

Table 7: Test error rate (%) on AdaAug-transfer and AdaAug-direct with different augmentation configurations.

| Task | Dataset | Simple | Random | AdaAug (w/o diversity) | AdaAug |
|------|---------|--------|--------|------------------------|--------|
| transfer | Oxford 102 Flowers | 5.87 | 4.65 | 3.70 | **3.63 $\pm$ 0.14** |
| | Oxford-IIIT Pets | 26.42 | 23.31 | 21.18 | **20.14 $\pm$0.12** |
| | FGVC Aircraft | 19.87 | 18.70 | 17.62 | **17.50 $\pm$ 0.19** |
| | Stanford Cars | 13.64 | 12.27 | 11.62 | **11.51 $\pm$ 0.21** |
| direct | CIFAR-10 | 5.3 | 3.8 | 3.7 | **3.6 $\pm$ 0.11** |
| | CIFAR-100 | 26.0 | 22.4 | 20.7 | **19.8 $\pm$ 0.08** |

## A.3 Fine-tuning of the diversity parameters

The following experiments study the sensitivity of the temperature $T$ and magnitude perturbation $\delta$ on the Flower dataset by changing the values of $T$ and $\delta$ around the default values used in our original experiments. The fine-tuning of the diversity parameters reduces the test-set error rate of AdaAug-transfer on the Flower dataset from 3.63 to 3.49.

## A.4 Efficiency of Policy Search

We compare the GPU hours needed to search the augmentation policy between different Automated Data Augmentation methods in Table 9. Among the baselines, AdaAug is more efficient than AutoAugment, PBA and Fast AutoAugment.

## A.5 More analysis on AdaAug policy learning

### A.5.1 Convergence of policy learning

Here, we provide some insights and empirical evidences for the convergence in policy training. In particular, Figure 4 shows the training and validation losses when learning the CIFAR-10 augmentation policy. The training and validation losses converge to some fixed values towards the end of training. In addition, we also visualize the change of the augmentation parameters ($p$ and $\mu$) in Figure 5. The magnitude parameters start at a smaller value and converge towards the end of training. For the augmentation probability, most of the candidates are stabilized after certain epochs while

Table 8: Test-set error rate (%) on AdaAug-transfer on Flower dataset with different values of $\delta$ and $T$.

| $\delta$ | 0.1 | 0.2 | 0.3 | 0.4 | 0.5 | $T$ | 0.1 | 0.2 | 0.3 | 0.4 | 0.5 |
|---|---|---|---|---|---|---|---|---|---|---|---|
| $T = 3.0$ | 3.80 | 3.81 | 3.63 | 3.49 | 3.67 | $\delta = 0.3$ | 4.16 | 3.92 | 3.63 | 3.92 | 4.28 |

Table 9: GPU hours needed to find the augmentation policy using AA, PBA, Fast AA, DADA, Faster AA, and `AdaAug`.

| | AA | PBA | Fast AA | DADA | Faster AA | `AdaAug` |
|---|---|---|---|---|---|---|
| CIFAR-10 | 5,000 | 5 | 3.5 | 0.1 | 0.23 | 2.89 |
| SVHN | 1,000 | 1 | 1.5 | 0.1 | 0.061 | 1.63 |

some others are updated more frequently. These are the evidences that show the convergence of our proposed policy training method. For a more thorough analysis for the policy convergence, we would like to leave it as the future work.

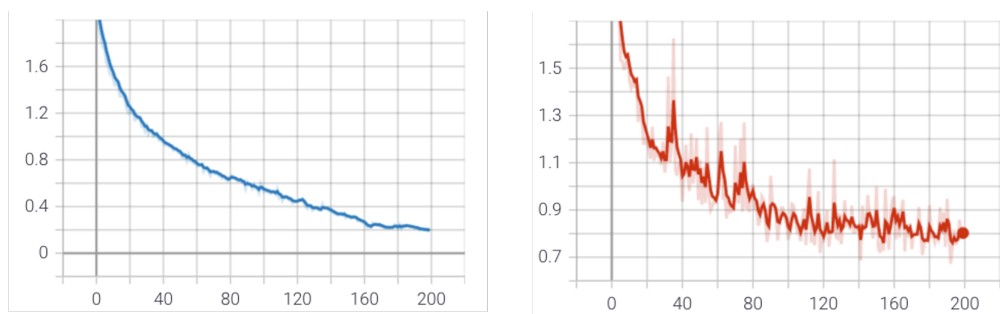

Figure 4: Convergence of the training loss (left) and validation loss (right) during the policy learning on CIFAR-10 dataset.

### A.5.2 ANALYSIS OF LEARNED AUGMENTATION MAGNITUDE

Complement with the augmentation probability in Figure 2, Figure 6 shows the augmentation magnitude for CIFAR-10 and SVHN. We observe that the policy magnitude $\lambda$ also shows slight variations among different classes and instances. Although the observation is less prominent and is harder to interpret when compared to the augmentation probability $p$, the learned augmentation magnitude adapts to different data samples.

### A.5.3 VARIANCE OF LEARNED AUGMENTATION POLICY GROUPED BY CLASS

In Figure 7, we plot the standard deviations of the predicted augmentation probabilities of the image instances grouped by its class label. It is observed that even within the same class, the predicted augmentation policy is different across instances. It is a clue that the `AdaAug` policy captures some instance-level augmentation information.

### A.5.4 DISTRIBUTIONS OF THE AUGMENTATION PARAMETERS IN ADAAUG-TRANSFER

In Figure 8, we show the distributions of the augmentation parameters when transferring the learned policy to the Flower, Pet, Car and Aircraft datasets. The distributions of the augmentation parameters show slight differences between different tasks. Apparently, we observe that the colour transformations, for example Colour, Invert and Solarize are less preferred in the Flower and Pet datasets, while shearing operations are relatively more favourable in the Car and Aircraft dataset. The observation of using less color transformation for the Flower dataset aligns with the findings from Lee et al. (2021). Although some differences in the augmentation distributions may be less

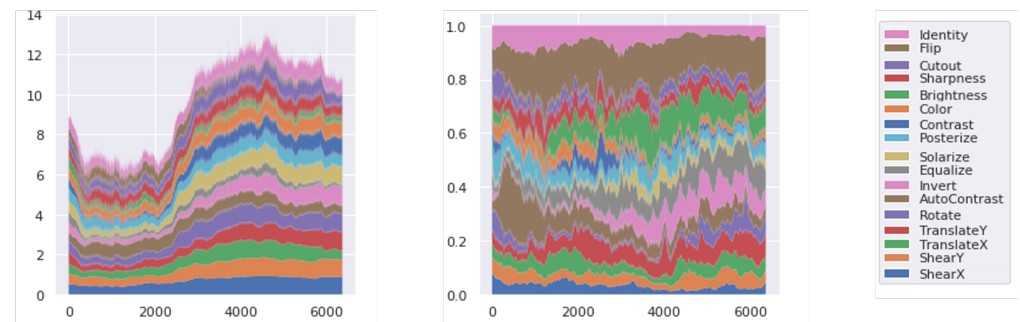

Figure 5: Update of the augmentation probability (left) and magnitude (right) during the policy learning.

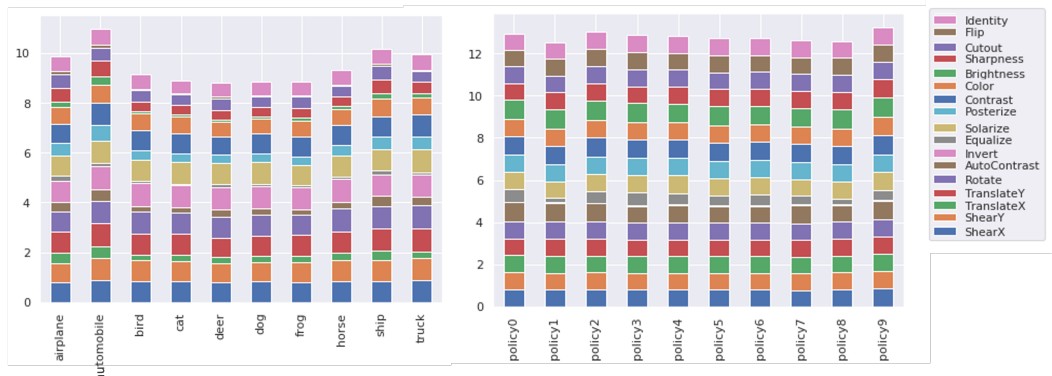

Figure 6: Illustration of the learned augmentation magnitudes for CIFAR-10 (left) and SVHN (right).

prominent and harder to interpret when compared to the AdaAug-direct cases, the policy shows its adaptations to different tasks.

### A.6 LIMITATIONS

Although `AdaAug` is differentiable and efficient in search, it requires forming different augmentation paths and passing the images through each path. Compared to standard training, where we forward $b$ images in one mini-batch, `AdaAug` processes $b \cdot |\mathbb{T}|$ images in the exploration pass. At inference time, `AdaAug` keeps a pre-trained policy network ($\pi_\theta$) to augment the new dataset. This adds extra computational effort when compared to other AutoDA policies, which may be a concern if the image resolution and model size are large but the computational resources are limited.

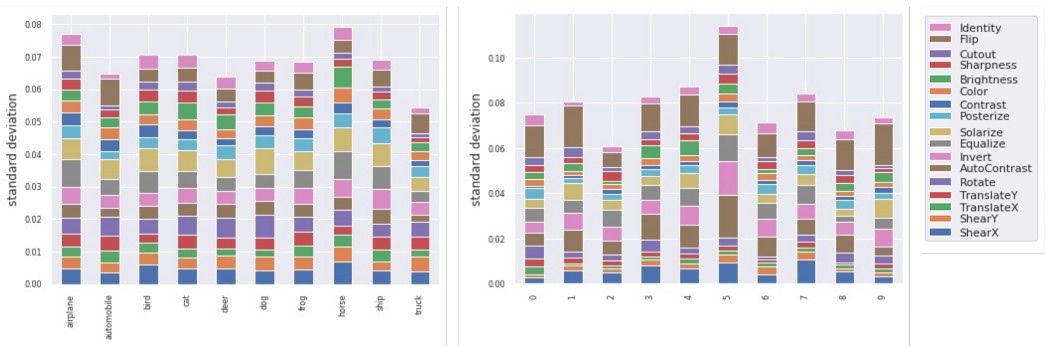

Figure 7: Illustration of the policy probability variations for CIFAR-10 (left) and SVHN (right).

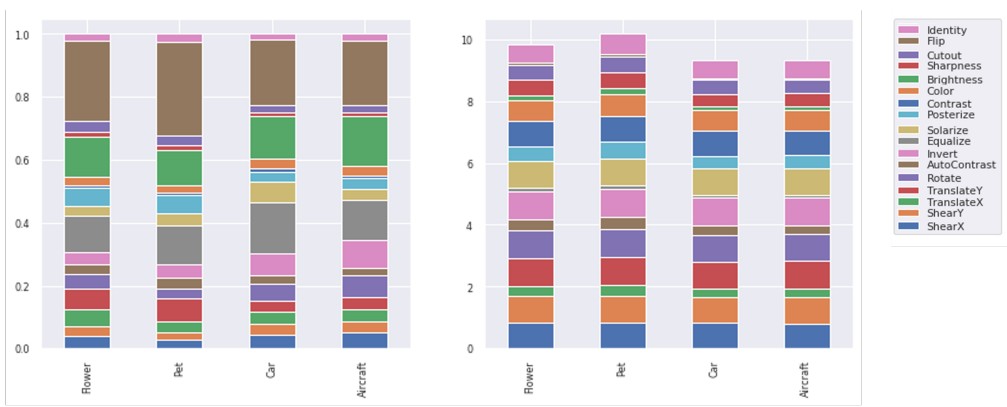

Figure 8: Illustration of the augmentation probability (left) and magnitude (right) when transferring the learned augmentation policy to the Flower, Pet, Car and Aircraft datasets.

