# OpenReview forum: "AdaAug: Learning Class- and Instance-adaptive Data Augmentation Policies"
_ICLR.cc/2022/Conference — ICLR 2022 Poster_

### Official Review · Reviewer_xMkp · 2021-11-02

**Correctness:** 3
**Technical Novelty And Significance:** 3
**Empirical Novelty And Significance:** 2
**Recommendation:** 6
**Confidence:** 4

**Details Of Ethics Concerns:**

None.

**Main Review:**

**Pros.**

- **Clarity**. Overall, the writing is clear and easy to follow
- **Well motivated problem and novel approach**. Instance-awareness of AutoDA is an essential but not explored aspect for better performance on the learned dataset and transferability to other datasets. The proposed idea that utilizes the hidden feature of the original input seems to be reasonable and novel.

**Cons.**

- **Experiments/Ablation for diversity parameters**. The authors introduce the three diversity parameters for further adapting the learned policy. However, the proposed method itself is also possible to adapt the augmentation based on the feature map of each instance. Why these additional diversity parameters are required? Is it necessary? Also, for a precise ablation, the results without these diversity parameters should be provided in a direct setup (Table2), and the comparison of distribution for resulting augmentations is required (e.g., with/without the diversity parameters, how the learned augmentation is adapted from CIFAR-100 to each target dataset?)
- **ImageNet experiments**.
    - First, it is quite confusing as the experimental results on ImageNet are presented in Section 4.1 (Transfer setting). Is it right or should it be moved to Section 4.2.?
    - Next, although the proposed method is quite worsen than other automatic augmentation learning methods in Table 2, I believe that the advantage of the proposed method is task-wise adaptability which more benefit from the transfer learning as done in Table 1 (as fine-tuning of the model from ImageNet pre-trained model is one of the most promising transfer learning nowadays) Can the proposed method outperform other fixed augmentation when we finetune the ImageNet pre-trained model?
- **Weighted summation within exploration**. According to Equation (2), the weighted summation of differently augmented samples is used to generate a prediction for updating augmentation policy. Hence, as the one sampled augmentation is only used at the exploitation, there is a discrepancy between exploration and exploitation in the way for using augmentation. Is the weighted summation is essential for exploration, or can it be replaced by another method? For example, one can use a Gumbel-softmax trick at exploration.

**Other comments.**

- **Illustration of the augmentation policy magnitude $\lambda$**. In Figure 2, the authors show the learned probability p to provide empirical evidence of class/instance-dependent augmentation. Since the magnitude $\lambda$ is also an important learnable parameter, it would be helpful to validate the proposed method if the corresponding illustration is additionally provided.
- **Qualitative results for better transferability with instance-awareness**. In Table 1, the proposed AdaAug significantly outperforms the other baselines by automatically adapting the augmentation. If the distribution of used augmentation can be compared qualitatively as done in Figure 2, it would enhance the empirical justification. (i.e., how the distribution of augmentation is **quantitatively** changed for the different tasks, instead of **qualitatively** as done in Figure 3?)

**Summary Of The Paper:**

- This paper proposes AdaAug, an Automated Data Augmentation (AutoDA) method to learn a class/instance-dependent augmentation policy efficiently. The key ideas of AdaAug are two-fold. First, it uses a hidden feature of the original input to adapt the augmentation for each instance. Second, it alternates exploit-and-explore procedures for efficiently updating the augmentation policy. The authors demonstrate the empirical effectiveness on two scenarios; 1) transfer and 2) direct. In transfer setup, AdaAug significantly outperforms other AutoDA baselines. In direct setup, AdaAug shows comparable performance with the baselines.

**Summary Of The Review:**

Although the target problem is important and the proposed method is novel, the empirical significances and analysis are quite insufficient currently. If the authors can provide that 1) experimental results fine-tuned from ImageNet model and 2) the empirical analysis how the augmentation parameters are actually changed by proposed method, especially in transfer scenario, I'll raise my score.

---

> ### Author Response · Authors · 2021-11-19
> **Thank you for your suggestions! The suggestions help to strengthen the empirical significance and analysis of our work.**
>
> __[C1] Ablation for diversity parameters.__ Thank you for your advice. As suggested by RandAugment (cited in the paper), larger datasets and more complicated models tend to perform better with stronger augmentations. Therefore we add the diversity parameters to create more variations from the original data. For example, if a 5-degree rotation augmentation is found to be effective for a given image, we would like to create other rotated versions of the same image. To clarify the effect of the diversity parameters, we add the ablation baseline AdaAug (w/o diversity) in Table 7 as suggested. It is observed that the learned adaptive policy itself can achieve good performance, while adding the diversity parameters can bring further improvements.
>
> Table 7: Test error rate (%) on AdaAug-transfer and AdaAug-direct with different augmentation configurations.
>
> | Task     | Dataset            | Simple | Random | AdaAug (w/o diversity) |      AdaAug      |
> |---|---|:---:|:---:|:---:|:---:|
> | transfer | Oxford 102 Flowers |  5.87  |  4.65  |   3.70   |  __3.63 $\pm$ 0.14__ |
> |          | Oxford-IIIT Pets   |  26.42 |  23.31 | 21.18 | __20.14 $\pm$ 0.12__ |
> |          | FGVC Aircraft      |  19.87 |  18.70 | 17.62  | __17.50 $\pm$ 0.19__ |
> |          | Stanford Cars      |  13.64 |  12.27 | 11.62  | __11.51 $\pm$ 0.21__ |
> | direct   | CIFAR-10       |   5.3  |   3.8  |  3.7 |  __3.6 $\pm$ 0.11__  |
> |          | CIFAR-100       |  26.0  |  22.4  |  20.  |  __19.8 $\pm$ 0.08__ |
>
> __[C2.1] ImageNet results.__ Yes, thanks for pointing it out. We think that moving it to Section 4.2 is more appropriate. We move the ImageNet results to Section 4.2 in the new version.
>
> __[C2.2] Fine-tune ImageNet pretrained model.__ Yes, we agree that the advantage of the proposed method is the task-wise adaptability. Thank you for the suggestion on adding the fine-tuning experiment. It definitely helps to strengthen our paper. In Table 5, we compare the performance when applying different AutoDA methods to fine-tune a ResNet-50 model pretrained on ImageNet for the transfer datasets. Similar to the AdaAug-transfer experiment, our method leads the other baselines in the fine-tuning scenario.
>
> Table 5: Test error rate (%) when fine-tuning pretrained ResNet-50 model on Oxford 102, Oxford 102 Flowers, Oxford-IIITPets, FGVC Aircraft, and Stanford Cars datasets.
>
> | Dataset            | Simple | AutoAugment | RandAugment | Fast AutoAugment |      AdaAug      |
> |--------|:------:|:--------:|:--------:|:----------:|:---------:|
> | Oxford 102 Flowers |  5.02  |     6.12    |     4.77    |       3.92   |  __2.81 $\pm$ 0.13__ |
> | Oxford-IIIT Pets   |  19.47 |    18.75    |    23.03    |   16.79   | __16.14 $\pm$ 0.07__ |
> | FGVC Aircraft      |  18.40 |    16.61    |    17.02    |   17.44   | __16.03 $\pm$ 0.15__ |
> | Stanford Cars      |  11.92 |     9.18    |    10.72    |   10.29    |  __8.82 $\pm$ 0.08__ |
>
> __[C3] Weighted summation within exploration.__ Yes, there is a slight discrepancy between the exploration and exploitation. We minimize the discrepancy effect by mixing the latent representations instead of the input images. Through mixing the representations in the latent space, $f$ receives non-mixed augmented images in both passes. Only $g$ receives non-mixed representation in exploitation and mixed representation in exploration. Regarding other exploration method, we think the Gumble-softmax trick can be used as an alternative to the weighted sum approach.
>
> __[O1] Illustration of the augmentation policy magnitude.__ Thank you for your suggestion. We observe that the policy magnitude $\lambda$ also shows slight variations among different classes and instances. Although the observation is less prominent and is harder to interpret when compared to the probability, we include the learned augmentation magnitude in Figure 6  for completeness of the study.
>
> __[O2] Qualitative results for better transferability with instance-awareness.__ We show the distributions of the augmentation parameters when transferring the learned policy to the Flower, Pet, Car and Aircraft datasets in Figure 8. The distribution of the augmentation parameters shows slight variations between different tasks. Apparently, we observe that the colour transformations, for example, Colour, Invert, Solarize are less preferred in the Flower and Pet dataset, while shearing operations are relatively more favourable in the Car and Aircraft dataset. Although such differences in the augmentation distributions may be less prominent and harder to interpret when compared to the AdaAug-direct cases, the policy shows its adaptations to different tasks.
>
> __[Summary]__ We thank the reviewer for the solid suggestions, which help to strengthen the empirical significance and analysis of our work. The experimental results fine-tuned from the ImageNet model is provided in Table 5. The empirical analysis of the change in the augmentation parameters is provided in the Appendix Section A 5.4 and Figure 8.

---

> > ### Comment · Reviewer_xMkp · 2021-11-22
> > **After rebuttal**
> >
> > Thank you very much for the response. I appreciate the effort that the authors put into addressing my questions. I believe that the above results and discussion can significantly improve the quality of the manuscript. My major concerns are mostly addressed; **hence I raise my score to 6 from 5**.
> >
> > However, I still think there is some room for improvement. For example, I recommend the following changes:
> > - Presenting Table 5 instead of Table 3. The main draft should consist of the most interesting results.
> > - Regarding Table 5, the corresponding parts (Section A.1.1) are not highlighted by colorization now (red).
> > - In Section A.5.4, the authors mentioned that "Apparently, we observe that the colour transformations, for example Colour, Invert and Solarize are less preferred in the Flower and Pet datasets, while shearing operations are relatively more favourable in the Car and Aircraft dataset." In [1], similar results are observed, and hence the citation of [1] might be helpful to enhance the results.
> >
> > [1] Lee et al., Improving Transferability of Representations via Augmentation-Aware Self-Supervision., NeurIPS 2021.

---

> > > ### Author Response · Authors · 2021-11-22
> > > **Thank you for your recommendations.**
> > >
> > > Thank you for your feedback. We have incorporated the changes in the new version of our paper.
> > > - Move the fine-tuning results to Section 4 and highlight the changes in red.
> > > - Include the suggested citation to enhance the results in Section A.5.4.

---

### Official Review · Reviewer_gmjs · 2021-11-07

**Correctness:** 4
**Technical Novelty And Significance:** 3
**Empirical Novelty And Significance:** 3
**Recommendation:** 8
**Confidence:** 5

**Main Review:**

**Strengths:**

- **S1.** **[reasonable and well-motivated]** The basic idea of this work that makes the augmentation conditional on data itself seems reasonable and worth exploring. Examples like "flip-invariance in digit classification" discussed in the paper motivates this work well.

- **S2.** **[relatively novel]** It is not a new idea to explore the class or subgroup conditional augmentations as introduced in the related work, but the originality may not be overwhelmed since this work may be the first trial to *automatically* learn instance-dependent augmentations.

- **S3.** **[enlightening]** The policy search phase (the exploration pass) in this method is elegant and extends the augmentation search space: it mixes the activations of different augmented images so that it avoids discretizing the augmentation magnitudes (while much prior work did this). This is not a major strength but a favorable bonus that allows better exploration for augmentation search, thus is appreciated.


**Weaknesses:**

- **W1.** **[missing ablation]** Some prior work like AutoAug [1] presents their search-free results to prove the augmentation policy does benefit from the search phase. If it is possible, these results should be shown to substantiate the benefits of the augmentation search. A straightforward way is to evaluate a randomly initialized $h_\gamma$ (not trained ever) on downstream experiments while keeping other *diversity parameters* the same.

- **W2.** **[unclear performance improvements]** The performance reported in Table 5. shows that large $\delta$ (*e.g.*, 0.4) would bring substantial improvements. The large-$\delta$ setting makes the augmentation policy close to a random one given the augmentation magnitude ranges from 0 to 1. This can raise the suspicion that the performance gains mainly come from the rich randomness of augmentation magnitudes, not the augmentation search algorithm. Reporting the variance (*e.g.*, standard deviation) of the augmentation magnitudes $\lambda$ and comparing it with the value of 0.4 would help further clarify this.


**Open questions:** (not weakness)

- **O1.** As a regularization technique, data augmentation almost always improves the loss, *i.e.*, augmented data have larger loss values than original ones. Thus, it is somewhat surprising that the augmentation policy does not collapse to "no augmentation" (*i.e.*, all magnitudes turn into zero) when we minimizing the loss of augmented data in the exploration pass. Could the authors explain this?


- **O2.** The authors claim that this method approximately reduces the distance between the densities of **augmented $D_{valid}$** and **$D_{train}$**. In contrast, I feel "reduces the distance between the densities of **augmented $D_{train}$** and **$D_{valid}$**" could be more natural and reasonable, as in most deep learning cases we apply data augmentation only on training data and hope these augmented samples could be like validation data.

    - In order to "reduce the distance between the densities of **augmented $D_{train}$** and **$D_{valid}$**", we need to use $D_{valid}$ to train $f_{\alpha}$ and $g_{\beta}$, and use augmented $D_{train}$ to optimize $h_{\gamma}$.

      - [use $D_{valid}$ to train $f_{\alpha}$ and $g_{\beta}$]: the exploitation pass does not perform any augmentations since we want to use $D_{valid}$ itself to train $f_{\alpha}$ and $g_{\beta}$. Thus, this pass could be understood as "encouraging $f_{\alpha}$ and $g_{\beta}$ to memorize the distribution of $D_{valid}$".

      - [use augmented $D_{train}$ to optimize $h_{\gamma}$]: the exploration pass could be regarded as "making augmented training data more close to validation data via minimizing the loss of augmented training data".

    - How do the authors think of this view?


-------------------

[1] Cubuk, Ekin D., et al. "Autoaugment: Learning augmentation strategies from data." Proceedings of the IEEE/CVF Conference on Computer Vision and Pattern Recognition. 2019.






**Summary Of The Paper:**

This paper proposes a search algorithm for instance conditional data augmentation. The contributions are two-fold: 1) it could be the first trial to automatically learn class-dependent or instance-dependent augmentations; 2) an efficient workflow is designed that not only realizes gradient-based augmentation search but also extends the search space of augmentation magnitudes to continuous.

**Summary Of The Review:**

I incline to give a marginal accept now given the originality and enlightening contributions. However, there are some questions towards the experiments that may affect my judgment. See main review for more details.

---

> ### Author Response · Authors · 2021-11-19
> **Thank you for your constructive feedback. The additional ablation study leads to a better analysis of the proposed model.**
>
> __[W1] Ablation on the random baseline.__ Using a “random” baseline to show the benefit of our search method is a good idea. We add the suggested experiments in Table 7, where the ‘Random’ baseline uses randomly initialized $h_\gamma$ while keeping the diversity parameters the same. The experiments show that the searched policy significantly improves the results over the ‘Random’ baseline.
>
> __[W2] Ablation on the diversity parameters.__ To clarify the performance improvements, we remove the diversity and evaluate AdaAug without the diversity parameters on different datasets. The ‘AdaAug (w/o diversity)’ baseline in Table 7 applies the learned augmentation policy to the target datasets without the diversity parameters. The study shows that the learned policy itself can achieve good performance, while adding the diversity parameters further improves the results.
>
> Table 7: Test error rate (%) on AdaAug-transfer and AdaAug-direct with different augmentation configurations.
>
> | Task     | Dataset            | Simple | Random | AdaAug (w/o diversity) |      AdaAug      |
> |----------|--------------------|:------:|:------:|:----------------------:|:----------------:|
> | transfer | Oxford 102 Flowers |  5.87  |  4.65  |          3.70          |  __3.63 $\pm$ 0.14__ |
> |          | Oxford-IIIT Pets   |  26.42 |  23.31 |          21.18         | __20.14 $\pm$ 0.12__ |
> |          | FGVC Aircraft      |  19.87 |  18.70 |          17.62         | __17.50 $\pm$ 0.19__ |
> |          | Stanford Cars      |  13.64 |  12.27 |          11.62         | __11.51 $\pm$ 0.21__ |
> | direct   | CIFAR-10           |   5.3  |   3.8  |           3.7          |  __3.6 $\pm$ 0.11__  |
> |          | CIFAR-100          |  26.0  |  22.4  |          20.7          |  __19.8 $\pm$ 0.08__ |
>
> __[O1] Policy collapsing to "no augmentation"?__ It is a good question. Empirically, we did not observe the policy collapsing to “no augmentation” in the experiments we conducted. Indeed, we find that the probability of applying ‘no augmentation’ gradually decreases with the policy training. We speculate that the classification network  $f \circ g$ gradually overfits to the training data, which may lead to a lower validation loss for the augmented samples. For example, if the unseen validation data contain some rotated versions of the training data, $f \circ g$ may perform better on the rotated validation images. Then, the policy is updated to predict a larger probability for the rotation augmentation.
>
> __[O2] Reduce the distance between the densities of augmented training data and validation data.__ It is an interesting suggestion to train the classification network on $D_{valid}$ and train the policy network on augmented $D_{train}$. We agree that the proposed idea may be more natural to some people. In terms of functionality, we think that both approaches approximately reduce the distance between the densities of seen data and unseen data, which helps to capture useful augmentation information.

---

> > ### Comment · Reviewer_gmjs · 2021-11-23
> > **After Rebuttal**
> >
> > Thanks for giving the additional ablation addressing my main concerns. The explanations about policy collapsing [O1] and a dual approach [O2] are also quite reasonable. The authors have added this extra ablation and analysis into their appendix, which could complement the existing ablations. Pretty interesting work. All in all: I slightly increase the score (to 8) and would like to see the paper published.

---

> > > ### Author Response · Authors · 2021-11-23
> > > **Thank you for the response.**
> > >
> > > Thank you for the positive feedback. We are pleased to hear that our response could address your concerns.

---

### Official Review · Reviewer_kS7y · 2021-11-09

**Correctness:** 4
**Technical Novelty And Significance:** 3
**Empirical Novelty And Significance:** 2
**Recommendation:** 6
**Confidence:** 4

**Main Review:**

**Strengths**
1. The motivation of this work is clear and the idea of searching data-dependent augmentation policies seems reasonable.
2. Empirical investigation is conducted on different scenarios.
3. The paper is well-written and easy to follow.

**Weaknesses**
1. There are some related works this paper omitted. For example, [1] has also considered individual sample variations in augmentation policy searching and it proposed a meta-learning approach to learn the augmentation policies. This paper can compare with this in terms of the idea and performance, so that we will know which approach is better under which scenarios. Also, regarding the performance, the authors mention AdaAug has achieved state-of-the-art performance on the CIFAR-10, CIFAR-100, and SVHN datasets. However, according to [1], it seems AdvAA [2] performs better than the baselines in this paper. Did the authors ever compare with this method? What are the results?
2. The performance improvement of the proposed approach seems marginal. Some prior works such as MetaAug [1] and AutoAug [3] run multiple times for the model and present the mean and standard deviation. Did the authors run multiple times for each model and what about presenting the mean and standard deviation etc?
3. Results on different neural network architectures are missed. Table 3 only presents results on the Wide-Resnet model. What is the performance on different models such as Shake-Shake (26 2x96d) and PyramidNet+ShakeDrop as shown in [1,2,3]?

**Other questions**
* As this work automatically learns the policy, we are curious about the issue of convergence during training. Is the convergence guaranteed on both training and validation data? Is it possible to do a theoretical convergence analysis?

**References**

[1] Zhou et al., MetaAugment: Sample-Aware Data Augmentation Policy Learning, AAAI 2021

[2] Zhang et al., Adversarial Autoaugment, ICLR 2020

[3] Cubuk et al., AutoAugment: Learning Augmentation Strategies from Data, CVPR 2019


**Summary Of The Paper:**

This paper introduces a data augmentation method AdaAug that learns adaptive augmentation policies in a class-dependent and potentially instance-dependent manner to improve the generalisation capability of deep learning models. Concretely, it proposes an efficient exploition-exploration workflow to search for an augmentation policy that optimizes the generalization performance. Experimental results on datasets with transfer and direct settings show the efficacy of this approach.

**Summary Of The Review:**

The topic of this paper is interesting and worth exploring. Overall, the paper is well-written and the idea is novel. However, there are some questions regarding the experiments and analyses. We would like to see the feedback of the authors to the questions above. Also, if the performance improvement of the proposed model is not significant, we would like to see some theoretical analyses on the advantages of this work.

---

> ### Author Response · Authors · 2021-11-19
> **Thank you for your constructive feedback. The discussions of additional related works, architecture-transfer and policy convergence have been added.**
>
> __[W1] Discussion of additional related works.__
> Thank you for mentioning MetaAugment and AdvAA  as the related works. We include the two methods in the related work section and discuss the criterion in selecting the comparison baselines under the experiment section. As a related work, MetaAugment learns an adaptive sample-weighting scheme and a global probability that controls the sampling of augmentation functions. It is different from our work, which learns a sample-specific augmentation parameterization (probability and magnitude). For AdvAA, it adopts an adversarial objective to learn the augmentation policy, while our method updates the policy to improve the validation performance directly. Admittedly, MetaAugment and AdvAA reported a slightly better performance in the direct experiment. However, our major focus is to study whether the instance-, class-adaptive augmentation parameterization is better than the non-adaptive approaches when transferring to larger or new unseen datasets. Therefore, we compare our method with AutoAugment, PBA, Fast AA, Faster AA, DADA and RandAugment, which learn the augmentation probability and magnitude parameters using a similar objective that improves the validation performance. We did not include MetaAugment and AdvAA in the comparison as some of their improvements may come from the re-weighting scheme and the adversarial objective. In the dataset-transfer scenario, based on our understanding, MetaAugment needs to relearn the weighting scheme and global probability parameter for new unseen datasets. We hope the mentioned reasons can justify our selection of the baseline methods and lead to a fairer comparison. We also think that the investigation of incorporating a learned weighting scheme and adversarial objective can be a future research direction to improve our method.
>
> __[W2] What about presenting the mean and standard deviation?__ The experiments are repeated on three random seeds. We report the average and state the standard deviations in the new version.
>
> __[W3] What is the performance on different models?__ Thank you for your advice to add the architecture-transfer experiment. As the major novelty of our approach is to learn a sample-adaptive augmentation policy, we put more efforts in testing the learned policy on unseen datasets under the dataset-transfer scenario (Table 1). However, we agree that studying the architecture-transfer would make our study more comprehensive. Due to the limited time, we conduct additional experiments on the smaller Reduced CIFAR-10 and reduced-SVHN datasets using Shake-Shake (26 2x96d). The results are reported in Table 6. In the two datasets we tested, the learned policy transfer well to the Shake-Shake (26 2x96) model.
>
> Table 6: Test error rates (%) of Shake-Shake (26 2x96d) on Reduced CIFAR-10 and reduced SVHN
>
> |                        | Simple | AutoAugment |  PBA  |      AdaAug      |
> |--------------|:------:|:-----------:|:-----:|:----------------:|
> | Reduced CIFAR-10       |        |             |       |                  |
> | Shake-Shake (26 2x96d) |  17.05 |    10.04    | 10.64 | 10.92 $\pm$ 0.07 |
> | Reduced SVHN           |        |             |       |                  |
> | Shake-Shake (26 2x96d) |  13.32 |     5.92    |  6.46 |  6.44 $\pm$ 0.17 |
>
> __[O1] Policy convergence.__ It is a great idea to study the policy convergence. Here, we provide some insights and empirical evidences on the policy convergence observed from our experiments. In particular, Figure 4 shows the training and validation losses when learning the CIFAR-10 policy. The training and validation losses converge to some fixed values towards the end of training. In addition, we also visualize the change of the augmentation parameters ($p$ and $\mu$) during policy training in Figure 5. The augmentation magnitudes start at a smaller value and converge towards the end of training. After some iterations, most of the augmentation probabilities are stabilized while some others are updated more frequently. These are the evidences that show the convergence of our proposed policy training. For a more thorough analysis of the convergence, we would like to leave it as the future work.
>
> __[Summary]__ We thank the reviewer for the in-depth comments and suggestions, which cover different important aspects in automated data augmentation.  In terms of the empirical significance of our work, we agree with the reviewer that our method slightly underperforms existing AutoDA methods in some experiments. However, we would like to emphasize that a larger focus (and also the novelty) of our approach is to apply a sample-adaptive augmentation policy to augment new unseen data, which is not studied in most of the existing works. Our method surpasses the performance of existing AutoDA methods in the AdaAug-transfer experiment. We also add new experiments to show the strength of AdaAug over other AutoDA methods when fine-tuning pretrained model on downstream tasks (see Table 5).

---

> > ### Comment · Reviewer_kS7y · 2021-11-23
> > **Appreciate The Authors' Responses**
> >
> > Thanks the authors for giving the explanation of baseline comparison, additional results, and policy convergence. I appreciate these responses, while I still have the following concern:
> > - The authors have explained the differences between the proposed model, MetaAug, and AdvAA. However, the advantages of the proposed method are not well-explained. Especially, MetaAug is also employing sample-aware data augmentation, which has similarities to the main novelty of the proposed method. Hence, I would suggest the authors to compare with this approach. If the improvement brought by the re-weighting scheme in MetaAug slightly outperforms the proposed method, is there any other aspect that the proposed method is better than MetaAug? For example, is the time complexity of the proposed method lower than MetaAug as we did not use the re-weighting scheme? Without a comprehensive explanation of the advantages of the proposed method, the contribution of this work might seem limited, as MetaAug has already investigated sample-adaptive augmentation. Moreover, since the proposed work uses class-adaptive augmentation which MetaAug did not investigated, will it be better than sample-adaptive augmentation? This paper lacks explanations of the differences between these two kinds of adaptive augmentation. Can we regard them as a similar mechanism? If the authors could give a reasonable explanation to the above concern, I am willing to increase my score.

---

> > > ### Author Response · Authors · 2021-11-24
> > > **Thank you for your feedback (2/2)**
> > >
> > > __4. Sample-adaptive versus class-adaptive.__ We show that AdaAug can learn class-adaptive and sample-adaptive augmentation, while MetaAugment only studied sample-adaptive augmentation. Here, we discuss two potential benefits of learning a class-adaptive augmentation. First, training the augmentation policy in a class-adaptive approach allows the data from the same class to share learned information and lead to faster convergence. Second, a class-adaptive augmentation policy is more interpretable, for example, the lower probability in flipping for some digits and the lower probability in colour transformation for flower classes. The investigation of a class-adaptive augmentation scheme helps to verify the approach through human inspection. However, it is rather difficult to explain and verify the sample-adaptive augmentation scheme predicted for each data sample.
> > >
> > > Moreover, it is an interesting question whether the sample-adaptive and class-adaptive augmentation can be regarded as a similar mechanism. In our implementation, we take the output from the last layer of a feature extraction network as the image representation to predict the augmentation parameters. As the image representation is close to the prediction target, we believe that the policy network can learn class-adaptive augmentation (as the sample within the same class would have a similar representation) and potentially some sample-adaptive augmentation. If we use or incorporate earlier layers to form the image representation, the learned augmentation can be more sample-adaptive. We think that the decision of using a sample-adaptive or class-adaptive augmentation depends on the actual task. A more in-depth study of this interesting issue may require more investigation and experiments, hence, we would like to leave it as the future work.
> > >
> > > ____
> > >
> > > We hope the discussions can address your concerns and better reveal the strengths of our proposed method. Please kindly let us know if you have any further questions or concerns.

---

> > > > ### Comment · Reviewer_kS7y · 2021-11-24
> > > > **After Rebuttal**
> > > >
> > > > Thanks the authors for giving the further explanation to address my concerns. The discussion about advantages of this work is reasonable and quite insightful. Hence I increase my score to 6.

---

> > > ### Author Response · Authors · 2021-11-24
> > > **Thank you for your feedback (1/2)**
> > >
> > > Thank you for your time and effort in reviewing our work. We discuss below the strengths of AdaAug over MetaAugment in terms of dataset transferability, technical difficulty, time complexity and class-adaptive capability:
> > >
> > > __1. Dataset Transferability.__ We believe the performance improvement in _dataset-transferability_ is one of the major advantages and contributions of our work over MetaAugment and other automated data augmentation methods. MetaAugment jointly trains the task network with an additional weighting network, which helps to re-weight the samples. Despite the good model performance, the learned weighting network is hard to be transferred to other unseen datasets. Specifically, the sample that is more important in a dataset (assigned with a larger weight by MetaAugment) is not necessary to be an important example in other datasets. Therefore, the weighting network has to be retrained for every dataset, which adds an extra burden when deploying the method to new tasks.
> > >
> > > On the contrary, AdaAug is designed to learn the mappings between an image and its effective augmentation, which allows the AdaAug policy to be transferred to other datasets easily. For instance, the flip operation found to be effective for the digit ‘0’ in a dataset is also an effective augmentation for the same digit in other datasets. As shown in the experiments, we can apply the learned AdaAug policy to new datasets without fine-tuning the policy network. To the best of our knowledge, such _transferrable_ sample- and class-adaptive augmentation scheme has not been studied by others in the research community. We understand that taking this approach may not be able to outperform some other dataset-specific method, like MetaAugment, in the direct case. However, we believe that dataset adaptability has a more general use case, which helps to save time and resources in solving new tasks and is more favourable to data practitioners.
> > >
> > > __2. Technical difficulty.__ Although both our work and MetaAugment use a sample-adaptive mechanism to improve the model performance, the learned policy from MetaAugment is different from ours, which results in a more challenging task to be solved. MetaAugment learns a single weighting for an augmented sample, while AdaAug learns a sample-adaptive augmentation parameterization, which has a much larger search space. Specifically, the search space of MetaAugment is $[0,1]$ and the search space of AdaAug is $([0,1] \times [0,1])^{|\mathbb{O}|}$, where $\mathbb{O}$ denotes the set of augmentation candidates. In addition, it is not trivial to apply the MetaAugment framework to learn the AdaAug augmentation policy as it involves sampling processes and non-differentiable operations. The mixing of the latent representations of the augmented validation data, which is a novel method for learning the augmentation probability, and the stop-gradient operation are some technical methods introduced in AdaAug to overcome the challenges.
> > >
> > > __3. Time complexity.__ Thank you for bringing up the discussion on time complexity. We notice that MetaAugment uses _three_ forward and backward passes in one iteration, while AdaAug uses at most _two_ forward and backward passes (one for training and one for validation). Moreover, we notice that MetaAugment searches the policy on the full dataset, compared to AdaAug which searches on a smaller subset of the dataset. Although MetaAugment did not report its exact computation time, we believe that AdaAug is a more efficient search method based on the above observations.

---

### Official Review · Reviewer_xQTc · 2021-11-14

**Correctness:** 4
**Technical Novelty And Significance:** 3
**Empirical Novelty And Significance:** 3
**Recommendation:** 6
**Confidence:** 4

**Main Review:**

Strengths

1. Adaptive augmentation by learning a class-dependent and potentially instance-dependent augmentation policy for each data instance is a nice idea with strong possibility in augmentation
2. Empirical studies has been conducted on different scenarios.
3. The paper is well-written and easy to follow.

Weaknesses
Not enough experiment. From the experimental results, it gives me an impression that the results are happen to be successful. Multiple trials with mean and variants should be given.

**Summary Of The Paper:**

This paper illustrates an adaptive based data augmentation method named as AdaAug that searches adaptive augmentation policies in a class-dependent and potentially instance-dependent manner to improve the generalisation capability of deep learning models. This paper proposes an efficient exploition-exploration workflow to search for an augmentation policy that optimizes the generalization performance. The empirical studies on some datasets shows that the performance increase with AdaAug.

**Summary Of The Review:**

I will give marginally acceptation for this paper. The idea is clear and novel but the novelty and contribution is not quite sounding. I can not decline that this is an interesting paper for this community.

---

> ### Author Response · Authors · 2021-11-19
> **Add more experiments and include the mean and variants of the results.**
>
> Thanks for pointing out the weaknesses of the experiment. Our experiments are repeated for three random seeds. We state the standard deviations in the new version.

---

### Author Response · Authors · 2021-11-19
**Summary of responses**

We thank all reviewers for the valuable comments. The suggestions can definitely help us to improve our work. Below please find a short summary of the major changes. The changes are highlighted in red in the new version of the paper.

__Related Work:__
-	We include two additional related works, MetaAugment and AdvAA, in Section 2.

__Experiments and Results:__
-	We add the criterion in selecting the comparison baselines under Section 4.
-	We add new experiments to evaluate our method when fine-tuning a pretrained model to four transfer datasets (see Table 5).
-	The experiments are repeated for three random seeds. We state the standard deviations of our method in the experiment section.
-	We move the ImageNet results to Section 4.2.
-	We add an experiment to study the use of AdaAug on a different network architecture in Table 6.

__Ablation Study:__
-	We conduct additional ablation study to clarify the effects of the searched policy and the diversity parameters in Table 7.

__Other Analysis:__
-	We provide empirical evidences to show the convergence of our policy learning method in Figure 4 and 5.
-	We add the illustrations of the learned augmentation magnitudes for CIFAR-10 and SVHN in Figure 6.
-	We analyze the distributions of the augmentation parameters across different downstream tasks to validate the adaptive power of AdaAug in Figure 8.

---

### Decision · Program_Chairs · 2022-01-20

**Decision:**

Accept (Poster)

**Comment:**

Reviewers agreed that this work is well-motivated and presents a novel approach for data augmentation around the adaptive augmentation policies. There were some concerns around the lack of ablation studies and unclear performance improvements, which were addressed well by the authors’ responses. Thus, I recommend an acceptance.